# Natural Substances as Valuable Alternative for Improving Conventional Antifungal Chemotherapy: Lights and Shadows

**DOI:** 10.3390/jof10050334

**Published:** 2024-05-05

**Authors:** Juan Carlos Argüelles, Ruth Sánchez-Fresneda, Alejandra Argüelles, Francisco Solano

**Affiliations:** 1Área de Microbiología, Facultad Biología, University Murcia, Campus Espinardo, 30100 Murcia, Spain; arguelle@um.es (J.C.A.); ruth.sanchez1@um.es (R.S.-F.); a.arguellesprieto@um.es (A.A.); 2Departamento Bioquímica, Biología Molecular B & Inmunología, Facultad Medicina, University Murcia, Campus El Palmar, 30112 Murcia, Spain

**Keywords:** antifungal agents, chemotherapy, natural products (NP), essential oils (EO), carnosic acid (CA), propolis (PP)

## Abstract

Fungi are eukaryotic organisms with relatively few pathogenic members dangerous for humans, usually acting as opportunistic infections. In the last decades, several life-threatening fungal infections have risen mostly associated with the worldwide extension of chronic diseases and immunosuppression. The available antifungal therapies cannot combat this challenge because the arsenal of compounds is scarce and displays low selective action, significant adverse effects, and increasing resistance. A growing isolation of outbreaks triggered by fungal species formerly considered innocuous is being recorded. From ancient times, natural substances harvested from plants have been applied to folk medicine and some of them recently emerged as promising antifungals. The most used are briefly revised herein. Combinations of chemotherapeutic drugs with natural products to obtain more efficient and gentle treatments are also revised. Nevertheless, considerable research work is still necessary before their clinical use can be generally accepted. Many natural products have a highly complex chemical composition, with the active principles still partially unknown. Here, we survey the field underlying lights and shadows of both groups. More studies involving clinical strains are necessary, but we illustrate this matter by discussing the potential clinical applications of combined carnosic acid plus propolis formulations.

## 1. Introduction

The present-day scenario of antimicrobial chemotherapy is extremely worrying. A recent report issued by the WHO alerts to the continuous increase in bacterial and fungal resistance to antibiotics in strains responsible for human septicemic infections [1]. Furthermore, the estimated elapsed time to have a new antibiotic approved by the sanitary services and applied in clinical therapies is 10–15 years. This long period usually discourages pharmaceutical companies from showing feasible interest in endeavoring new research screenings. In the last 7 years, only 12 new compounds have been authorized for clinical use [1]. The search for antibacterial agents has always deserved preferential interest from health services worldwide. A recent survey indicates that in 2022, 64 antimicrobial drugs were subjected to clinical assays, of which only 8 were developed by big companies, and just 1 (fosravuconazol) was approved as a new antifungal agent, for mycetoma [2].

Antifungal chemotherapy has remained scarce since the number of pathogenic fungi has traditionally been lower than bacteria and viruses, and they are usually sorted as secondary opportunistic pathogens after a primary bacterial or viral infection [3,4]. Except for particular cases and complications, primary mycotic infections are considered superficial, with a good prognosis even in the absence of any treatment [5,6]. However, this view has dramatically changed during the past few decades, principally due to the growing of immunosuppressed populations suffering chronic diseases susceptible to the increasing spread of systemic fungal infections, with the subsequent sharp rise in morbidity and mortality rates [4,7,8]. Thus, the annual mortality associated with fungal infections accounts for roughly 1.6 million [9]. The majority are caused by a few fungal genera, mostly *Candida*, *Cryptococcus*, and *Aspergillus*, whereas *Pneumocystis*, *Histoplasma*, or *Mucor* seem to have a minor incidence [9]. *Candida albicans* remains the most prevalent human fungal pathogen and is also considered the fourth cause of nosocomial infections [8], but other “non-albicans” species of *Candida*, initially cataloged as innocuous, are more and more progressively isolated as being responsible for dangerous outbreaks in hospitals [10]. The incidence of candidemia presents significant variations worldwide, with an estimated range of 1–10 per 100,000 persons. This setting is even more hazardous after the appearance of *C. auris*, a highly transmissible multiresistant yeast, cataloged as an “urgent threat” by the Centers for Disease Control and Prevention (CDC) [9,10].

The arsenal of chemotherapeutics currently available is restricted and not sufficient to combat emerging fungal infections. A limiting factor comes from the fact that fungi are eukaryotic organisms. Therefore, they share a similar cellular organization with their hosts, lowering the specific therapeutic cell targets available and reducing the selective toxicity among infective fungal and host cells [4,11,12]. Likewise, the increasing isolation of so far harmless fungi involved in severe outbreaks, mainly recorded among the immunocompromised population, together with the rising extension of fungal resistance to customary antifungals, also makes the establishment of standard protocols for safe and effective treatments against septicemic mycosis very difficult [4,5,12,13]. Therefore, the discovery of novel antifungal drugs is extremely challenging, as well as the subsequent elaboration of culture conditions for microorganisms able to synthesize feasible new hopeful drugs, or the processing methods for their reliable purification and standardization, as reviewed elsewhere [14].

Thus, other emerging treatment options and novel formulations are needed to improve overall drug efficiency. Due to the restrictions of conventional antimycotic therapies, our aim herein is to focus on a distinct type of compound that can be an alternative for the replacement or reinforcement of antifungal chemotherapy. These sorts of products are natural substances, which have been proposed as an appropriate and useful complementary choice in antifungal therapy, particularly for counteracting resistance. They are usually extracted from medicinal plants and have been applied worldwide as remedies in the pursuit of human well-being since ancient times. They may play an important role in alleviating infectious diseases, neutralizing fungal pathogens by a variety of mechanisms, and being generally different from those involved in chemotherapy [15].

In the above-referred dual context, the main goal of this review is to provide a broad outlook on new antifungal approaches, including an evaluation of the potential improvement in antifungal chemotherapy by means of experimental combinations among the conventional compounds clinically available and natural products (NPs, hereinafter). The approaches so far tested look for synergic effects and improvement in the antifungal-induced damage in hosts. However, in some cases, this kind of treatment displays unforeseen interferences that should be discussed. This contribution does not aim to present a complete and exhaustive survey of structures, mechanisms, and properties displayed by the classic antifungals or the bioactive compounds present in NPs, or the newly discovered molecules with promising antifungal activity. For this purpose, there are several excellent and updated reviews published in recent years that can be easily consulted, since the majority are accessible in an open-access format [9,11,14,16,17,18,19].

## 2. Current Clinical Chemotherapies against Fungi: Main Families and Some Drawbacks

A key bottleneck regarding present-day antifungal therapy is the relatively small amount of truly active and safe compounds applied in medical practice. They can be classified according to their mechanism of action and chemical structure. Three main families of antifungals are currently available: polyenes, azoles, and echinocandins [8,11,12]. A scheme of their specific cell targets and basic mechanisms of action is shown in Figure 1. The earliest agents, polyenes, were introduced 60 years ago as unmodified molecules obtained from natural sources, but their fungicidal activity was later improved by chemical modifications, giving rise to semisynthetic compounds more effective than the initial precursors. However, their administration may yet generally provoke some undesirable toxic effects [3,9,11,16,20].

Among polyenes, the macrolide Amphotericin B (AmB) is still the most widely used as a fungicide against both superficial and systemic candidiasis, as well as other mycoses [11,16,17]. Polyenes bind to ergosterol, the essential sterol present in the fungal membrane, triggering alterations in its selective permeability that can eventually lead to cell death [9,16]. The effect is preferential but not specific to ergosterol, so that it can also weakly bind to sterols present in the human cell membrane as a secondary target, causing a certain loss of permeability and metabolic alterations in the host. Due to this undesirable side effect, AmB causes extensive hemolysis and toxicity in the liver and kidneys [21]. These adverse effects have largely been circumvented by new liposomal formulations, although their prescription has become expensive [4,22]. In turn, other introduced members (i.e., nystatin or natamycin) belonging to this family display higher toxicity [9] and need long-lasting treatments, inconvenient for chronic infections. On the other hand, cases reporting clinical resistance to AmB and other polyenes remain surprisingly rare 60 years after its first application as monotherapy [11,20]. For this reason, it is still customarily employed to combat severe mycoses.

Azoles are organic molecules extensively used in antifungal therapy [23]. The triazoles of second generation (i.e., voriconazole or posaconazole) exhibit a wide spectrum of activity against opportunistic yeasts, and they have been approved for the prophylaxis of invasive infections caused by *Candida* and *Aspergillus*. A major trouble is that azoles act essentially as fungistatic rather than fungicides, which might induce the appearance of resistant strains, although the newly approved compounds display higher fungicidal action with a broader spectrum of activity [9,16,17,23]. Azoles interfere with the ergosterol biosynthesis from lanosterol by inhibiting specific steps of the pathway, causing an ergosterol deficit, which impairs the correct structure of the cell membrane. Azoles inhibit the first and key enzyme of this pathway, referred to as Erg11p, lanosterol 14α-demethylase, or Cyp51. This enzyme belongs to the cytochrome P450 monooxygenase superfamily, specifically the mammalian Cyp51 isoenzyme [24,25]. In addition to this target, some recent azoles (i.e., voriconazole) also inhibit a sterol-24-C-methyl-transferase encoded by *ERG6* involved in an ulterior step of ergosterol synthesis, increasing its efficiency in comparison to the primary imidazolic azoles. It is worth noting that an ortholog of the fungal Cyp51 involved in animal sterol metabolism is also present in human cells. The dual target of both Cyp51 isoenzymes is responsible for the adverse effects provoked by azole chemotherapy. Azolic treatments cause liver dysfunction (i.e., ketoconazole), and skin edema with irritation, rash, and itching in the mucosa membranes. For other azolic drugs, such as miconazole, a deficient absorption in the digestive tract has been reported [9]. 

Resistance to azole compounds has increased in recent years due to frequent spontaneous mutations recorded in the gene encoding Cyp51 (*ERG11*), as well as in genes involved in the drug efflux pumps (*CDR1*, *CDR2*, and *MDR1*) that allow the mutants an efficient removal of the antifungal drugs from their internal milieu [16,26]. The formation of compact active biofilms or alterations of metabolic pathways, leading to a reduction in or loss of function, are also additional ways to acquire resistance. 

The third main family involved in antifungal chemotherapy corresponds to echinocandins (caspofungin, micafungin, and anidulafungin) [27,28]. These compounds are semisynthetic lipopeptides, whose composite structure consists of a cyclic hexapeptide core plus a specific hydrophobic side chain implied in the antifungal activity. Echinocandins act as non-competitive inhibitors of the β-(1,3)-D-glucan synthase that catalyzes the formation of β-glucan polymers, essential components of the fungal cell wall [17]. However, the susceptibility of different fungal species to echinocandins shows ample diversity. These agents display fungicidal activity against many clinical strains of *Candida*, but they behave mainly as fungistatic against *Aspergillus* spp. Notably, the majority of *Cryptococcus* strains are refractory to echinocandins [16,22,26,29]. In addition, some opportunistic yeasts containing a mutated *FKS1* gene (encoding one subunit of the glucan synthase complex) show resistance to these compounds [26]. Furthermore, although echinocandins are directed against cell wall formation, a specific target of fungal cells, these chemicals trigger some side-off gastrointestinal and cardiac adverse effects in treated patients [30], which limits their utility as antifungals with a wide spectrum.

Besides those undesirable effects derived from their intrinsic relative toxicity and the appearance of fungal resistance, other recently unveiled factors increase the complications for well-defined, reliable, and safe treatments. For the three families, the disclosed mechanisms of action focused on either the plasma membrane and/or fungal cell wall as main targets (summarized in Figure 1). However, accumulated strong evidence suggests a more pleotropic mode of action than initially thought. Mitochondrial and cytosolic oxidative damage is also involved in fungal cytotoxicity exerted by most of the antifungal drugs [31]. The increased formation of reactive oxygen species (ROS) seems to be a general effect of AmB on pathogenic yeasts [32], although it is not merely caused by alterations to membrane permeability, since the AmB-induced oxidative stress in *C. albicans* is mediated by the Hog1 signaling pathway [33]. Similarly to that of polyene, the fungicidal action of echinocandins, such as micafungin, is also accompanied by an increase in intracellular ROS formation [34]. Concerning azoles, the evidence in favor of internal oxidative stress is also consistent, although the mechanism remains partially unknown. Available data support itraconazole-induced ROS production in *A. fumigatus* and *C. parapsilosis* [35,36].

In addition to the three main families considered above, other antifungal agents endowed with different chemical structures and mechanisms of action have been eventually applied for the treatment of some particular mycoses (i.e., 5-fluorocytosine, terbinafine, ciclopirox, and griseofulvin derivatives have recently been reported [37]). Griseofulvin is a natural compound isolated from *Penicillium griseofulvum*, but all of them are chemically synthetic drugs able to inhibit both cell growth and cell division, either by interfering with the replication and transcription of nucleic acids or the microtubule dynamics needed for the formation of the mitotic spindle (Figure 1). Obviously, these chemicals also cause severe cell damage and undesirable effects in infected patients since they have analog targets in the human nuclei and microtubules. Some recent and very detailed reviews about chemotherapy based on these molecules are available [9,11,17,31].

In summary, the evidence collated points to the urgent necessity to search for new molecules endowed with more potent and safer antifungal activity, as well as to improve the efficiency of the available compounds by either reducing their unwanted effects or preventing the appearance of resistant strains. Although their structural components and key metabolic pathways must be the preferential fungal targets, this approach has already been tried for decades, limiting the scope. Currently, some molecules are in distinct phases of clinical trials designed for the treatment of invasive mycosis [14,17,38,39], which could improve the future scenario. 

## 3. Natural Products: Potential Adjuvants of Antifungal Chemotherapy

An NP is a very wide and ill-limited term that typically refers to substances derived from any cellular source, such as bacteria, fungi, plants, sponges, and other marine organisms and animals, encompassing insects (i.e., bee propolis) to higher mammals (i.e., musk from male deer). For centuries, this large number of natural extracts has been used for various purposes, including food, cosmetics, industrial, and medical applications. In this revision, we refer to NPs from a concrete point of view, restricted mostly to natural extracts obtained from plants showing beneficial effects on human health, including antimicrobial action. Plants are rich sources of bioactive secondary metabolites of a wide variety, such as alkaloids, carotenoids, flavonoids, polyphenols, saponins, tannins, terpenes, and terpenoids, among other bioactive compounds [40]. 

These molecules have customarily been involved in antioxidant, anti-inflammatory, antitumor, antiviral, antibacterial and antifungal actions. Here, we will focus on the antifungal effect whether individually or in combination with the previously described chemotherapeutic agents. More than one hundred of those NPs have been characterized as antifungals. Perhaps prematurely, this activity has been assayed against relevant pathogenic fungi under laboratory conditions, but the results were frequently extrapolated to general ones. 

Before going into a general survey on NPs, it is worth making clear that the unwanted effects caused by the extracts in the recipient host are normally lower and weaker than those produced by the clinical agents, but their antifungal activity is also weaker. Although, for some of them, the antifungal properties are comparable to chemotherapeutical drugs described in the previous section, they are far from being the first choice for use in clinical practice yet. Some NPs are used as substitutes for conventional antifungal therapy, but the data indicate that they should be introduced as effective principal antifungals only in certain situations. For instance, they might be useful for the treatment of some *Candida* spp. infections that are becoming increasingly resistant to first-line and second-line antifungals, such as fluconazole and echinocandins. Fluconazole-resistant “non-albicans” *Candida* has emerged as major cause of candidemia. NPs might be an alternative or, at least, a complement in these cases. Concerning this aim, we will mainly focus on the current information about these compounds as well as on several trials developed for testing their antifungal capacity, including our experience with two of those natural agents: a semipurified extract of rosemary leaves rich in carnosic acid, and hydroalcoholic propolis extract [41,42].

A significant number of NPs are usually reported as essential oils (EOs), which are complex mixtures of volatile molecules, usually monoterpenes, sesquiterpenes, and low molecular weight oxygenated derivatives [43]. These oils have been widely applied in food preservation, but scarcely as direct antifungals. The most common mechanism of action is related to their capacity to create lipophilic chaos, with disruption of cell wall and plasma membrane integrity, altering the permeability of fungal cells to ions. They also induce vacuolation (Figure 1), damage to cellular organelles, and disorganization of hyphal cytoplasm, especially during the transition phase. Although their action is usually fungistatic rather than fungicidal, at high concentrations, complete cell autolysis can be achieved [44].

In addition, some EOs have evolved other specific mechanisms of action, at least partially responsible for their antifungal activity. To briefly describe some illustrative examples, similarly to azoles, clove oil inhibits ergosterol synthesis since it contains eugenol, a potent Cyp51 inhibitor [45]. Oils from oregano, lemon grass, thyme, clove, cinnamon, and peppermint inhibit both planktonic growth and biofilm formation of cryptococcal cells, disrupting their quorum sensing and survival [45]. The antibiofilm activity of the tested active components was in the following order: thymol > carvacrol > citral > eugenol = cinnamaldehyde > menthol. Different mechanisms of action have been proposed, which include Ca(II) and endoplasmic reticulum stress, induction of the unfolded protein response, and damage to cell systems involved in energy production, in addition to blocking ergosterol biosynthesis and membrane disruption [46]. Those actions are not constrained to the most active monoterpenes such as thymol and carvacrol, since several EOs also contain different bioactive compounds, such as the antioxidant flavonoids. Thus, apigenin and quercetin induce the downregulation of the *ERG11* gene, encoding the Cyp51 enzyme [47,48]. Notably, the same studies also indicate that flavonoids enhance *CDR1* and *CDR2* expression. These genes encode drug efflux pumps, so that their antifungal effects can be counteracted [48]. However, the effect on CDR expression could be the opposite. Garlic contains a peptide that causes downregulation of the CDR1 gene, and the drug resistance almost disappears after 21 days of serial passages of *C. albicans* cultures in the presence of garlic oil [49]. In addition to this novel peptide, the well-known allicin and other organosulfur-related compounds found in garlic and onion appear to inactivate key thiol-dependent enzymes [50]. Finally, it has also been described that several bioactive compounds contained in EOs have immune-stimulating properties, exerting their antifungal beneficial action by favoring the human immune response against fungal infections rather than by a direct fungicidal activity [51].

Table 1 contains, in alphabetical order, the most used NPs, associated with their most important bioactive component(s), and some references for specific details.

3.1. Aloe vera. This is a well-known medicinal plant used for many therapeutic purposes. The ethanolic extract of Aloe vera leaves shows antimicrobial properties against different strains of bacteria and pathogenic fungal strains [52], and Aloe vera gel has been effective against oral candidiasis. However, the efficacy of this gel was poor in comparison to nystatin, so the inhibitory concentrations of extracts should be higher than 100 mg/mL [53].

3.2. Barberry. This extract is one of the more studied natural herbal remedies for its ample use in alternative and complementary medicine. Its beneficial action is due to its high content in berberine, a quaternary ammonia alkaloid [54], which shows strong toxicity against *Candida* spp. with a MIC range as low as 10 µg/mL. This molecule is a direct inhibitor of the Cyp51 isozyme, affecting ergosterol synthesis [55]. In fact, berberine is found in many plants, and it also acts on several enzymatic activities, whose detailed description is out of the purpose of this review.

3.3. Pepper: Black pepper oil is applied as a natural preservative, although it also possesses antioxidant, hepatoprotective, and antifungal activities [56,57]. It possesses the capability of changing membrane permeability and causes mitochondrial dysfunction in *Aspergillus flavus*, *Aspergillus ochraceus*, *Fusarium graminearum*, and *Penicillium viridcatum*. The nature of the bioactive molecules involved in its antifungal action is unclear. According to Singh et al. [56], the oil is rich in mono and sesquiterpenes (β-caryophyllene and limonene), but Zhang et al. [57] suggested that phenols, flavonoids, and proanthocyanidins are predominant in this oil. Moreover, the acetone extracts are rich in alkaloids (piperine and others). Hence, in summary, the composition is complex, and the published reports are focused on different types of molecules.

Related to black pepper, fungicidal effects have also been described in cayenne pepper (*Capsicum frutescens*). These extracts, besides the capsaicinoids responsible for its counterirritant and appetite stimulator activities, contain CAY-1, a novel saponin with antifungal activity [58]. CAY-1 appears to act by disrupting the integrity of fungal membranes, as reported from 16 different species, including *Candida* spp. and *Aspergillus fumigatus*, with a MIC ranging from 4 to 16 µg/mL. It was especially active against *Cryptococcus neoformans* (90% inhibition at 1 µg/mL). Synergistic activity against *C. albicans* and *A. fumigatus* was also observed between CAY-1 and AmB. Interestingly, Renault et al. [58] reported that no significant cytotoxicity was observed when CAY- 1 was tested against 55 mammalian cell lines at 100 µg/mL.

3.4. Cinnammon: This oil is a well-known effective antimycotic agent [59], essentially due to its content in cinnamaldehyde. This agent shows some inhibitory action on Cyp51 and also increases ROS formation. At MICs relatively low (i.e., 25 µg/mL), it damaged the integrity of both cell wall and plasma membrane and disrupted the redox cellular homeostasis of *Aspergillus niger* [60]. In addition to its direct antifungal features, cinnamon significantly reduced the damage caused by *C. albicans* and *C. auris* on red blood cells, and it may be used as an alternative treatment for patients with candidiasis [61].

3.5. Clove: This oil has considerable antifungal activity against clinically relevant fungi, including fluconazole-resistant strains [62] and *Candida auris* [63]. This oil contains a high percentage of eugenol, a phenol triggering a significant reduction in ergosterol synthesis. It has recently been reported that eugenol and its derivatives are able to inhibit Cyp51 activity and the synthesis of this fungal-membrane-specific sterol, resembling the azole effect [44]. As a consequence, the germ-tube formation in *C. albicans* was almost completely inhibited by both the whole oil and purified eugenol at MIC values relatively low. In fact, clove and cinnamon oils are included as the most potent antifungal agents so far obtained from medicinal herbs [61].

3.6. Cumin: This seed oil shows significant toxicity when assessed against 75 clinical isolates of *C. albicans* (MIC about 0.3–0.6 µL/mL) and non-albicans *Candida* spp. (MIC of 0.02–1.25 µL/mL (*v*/*v*)). Recently, aqueous and methanol extracts from plant seeds have proven to have antifungal effects on fluconazole-resistant *C. albicans* [64]. Resistance to antifungal drugs in these opportunistic yeasts has increased in the last years by several factors, such as biofilm formation and overexpression of genes for efflux drug pump, contributing to its great pathogenic potential. Thus, cumin seed oil is an effective natural anticandidal agent with good prospects for the treatment of superficial and mucosal candidiasis, including vaginal candidiasis [65].

3.7. Garlic: This is a well-known traditional plant endowed with antimicrobial activity. Garlic extract has universally been used in folk medicine with multiple beneficial effects, including natural antimicrobial activity, with both antifungal and antibacterial features. The extract contains more than 200 phytochemicals, including allicin and other related organosulfurs, the specific active principles of freshly crushed garlic [66], whose antimicrobial action is due to the reaction with thiol groups of various essential fungal enzymes, including thioredoxin reductase, RNA polymerase and DNA gyrase, which inactivates those activities and impairs, on the one hand, intracellular redox balance with membrane lipid peroxidation and, on the other hand, transcription and DNA replication, respectively [66]. In vitro studies have shown that garlic has a strong antifungal activity against various species of *Candida* including both fluconazole-susceptible and fluconazole-resistant strains of *C. albicans* [49,67]. Notably, allicin enhances the toxic effect of AmB and fluconazole against these species [66,68].

It has been also reported that allicin and possibly other organosulfur compounds also act by inhibiting the quorum-sensing activity of fungal cells [69]. Pure garlic showed a marked antifungal efficacy when compared to onion and lemon juice extracts against *C. albicans* [47,67,69,70]. This indicates that allicin is more potent than limonene and other terpenes present in lemon extracts, and that garlic has a higher allicin content than onion. Furthermore, it must be remarked that garlic effects are not exclusively based on allicin. A new nonapeptide, called NpRS, with antifungal activity and low toxicity for mammalian cells, has recently been isolated from garlic. This molecule is a kind of cationic cycled peptide that causes membrane disruption, ribosome interference, and downregulation of the gene *CDR1*. In the presence of this peptide, the resistance to azoles was remarkably attenuated after 21 days of serial passage [49]. 

3.8. Mānuka: This NP is presented asoil or honey. It is obtained from *Leptospermum scoparium*, a plant widely consumed by the indigenous populations of New Zealand and Australia for centuries. Evidence about the antifungal, anti-parasitic, and anti-inflammatory activities of mānuka oil, its analogs, and its components has been reviewed [71]. Similar preparations, such as Agastache honey, showed higher in vitro antifungal activity than mānuka against *Trychophyton mentagrophytes*, *T. rubrum*, and *C. albicans* [72]. In both cases, the effect was weak and fungistatic rather than fungicidal. As with other oils, authors advise that more clinical evidence on its efficacy, safety, and dosing guidelines is still necessary before their routine application against epidermic superficial fungal infections. Concerning composition, the main bioactive identified compounds in mānuka were acetanisole and methyl-3,5-dimethoxybenzoate, but phenol, 2,4-bis(1,1-dimethylethyl) and estragole are present in Agastache. 

3.9. Maqui Berry: This extract is obtained from the *Aristotelia chilensis* fruits (Molina, Stuntz), a plant traditionally employed in the ethnomedicine of Chile and Patagonia, very rich in anthocyanins [73]. This NP inhibits filamentation of *C. albicans*, in both laboratory and clinical strains. The morphogenetic yeast-to-hyphae transition is a key virulence feature for the establishment of local and systemic *C. albicans* infections. The extract acts synergistically with nystatin, the combination of ¼ × MIC of nystatin and 125 mg/mL of the extract completely prevented filament formation, without any toxic effect [74].

3.10. Neem: Theseleaf extracts are obtained with water, ethanol, and ethyl acetate. All of them inhibit in vitro the growth of certain human pathogen fungi (*Aspergillus flavus*, *A. fumigatus*, *A. niger*, *A. terreus*, *C. albicans*, and *Microsporum gypseum*) [75,76], although the efficiency of the preparations is moderate, and they should be used at relatively high concentrations. A 20% ethyl acetate extract produced the strongest inhibition. A main component in these extracts is azadirachtin, a highly oxygenated triterpenoid used as a biodegradable insecticide that shows antifungal activity. Looking for new bioactive compounds, HPLC analysis of the last extract revealed the presence of a new component (nimonol), but purified nimonol, as a separate compound, did not show any antifungal activity when assayed against any of the six strains [76].

3.11. Olive: This oil shows conspicuous antifungal in vitro activity against *Candida* isolates from the bloodstream. Some “non-*albicans*” *Candida* species have recently emerged as a major cause of systemic bloodstream infections. These isolates display increasing resistance to first- and second-line antifungals, such as echinocandins and fluconazole. Interestingly, both olive and cinnamon oils display marked sensitivity against both *C. albicans* and non-*albicans* spp. [77]. They were found to be effective against nearly 50% of the *C. albicans* and 55.5% of fluconazole-resistant *C. krusei* isolates. In turn, ozonated olive oil displayed even higher capacity to inhibit growth of *C. albicans*, *C. glabrata*, *C. krusei*, *C. parapsilosis*, and *Saprochaete capitata* [78], so that natural or better ozonated olive oil may help to control both fluconazole-resistant and -sensitive fungal strains.

3.12. Oregano: This oil also contains two bioactive molecules, carvacrol and thymol, with antifungal activity. Those terpenes are also found in other EOs, such as those obtained from pepperwort, wild bergamot, and rosemary, with similar activity to oregano. Their cell toxicity has been demonstrated against *C. albicans*, individually and in combination with fluconazole or AmB [79,80]. 

3.13. Rosemary: The inhibitory effect of rosemary (*Rosmarinus officinalis*) oil was tested against *Alternaria alternata*, *Botrytis cinerea*, and *Fusarium oxysporum* [81]. This essential oil mostly contains monoterpenes, oxygenated monoterpenes, and sesquiterpenes, but phenols such as chlorogenic acid and flavonoids are also present [82]. The main constituents of the oils were p-cymene (44%), linalool (20.5%), and γ-terpinene (16.6%). Other studies analyzed the antibiofilm effect of *R. officinalis* L. extract on *C. albicans*, *C. dubliniensis*, *C. glabrata*, *C. krusei*, and *C. tropicalis*. The inhibition degree of fungal growth depends on the dose. Rosemary extract showed an antibiofilm effect on *Candida* spp. comparable to nystatin, although at higher doses (50 to 200 mg/mL). 

3.14. Tea tree: This oil is one of the most common herbal remedies due to its anti-inflammatory and antifungal activity, being particularly rich in beneficial mono- and sesquiterpenes and their respective alcohols. It is composed of approximately 100 substances, predominantly monoterpenes (terpinolene, α-pinene, 1,8-cineole, p-cymene, γ-terpinene, terpinol, and terpinen-4-ol) [83]. The main antifungal action of both the oil and some of its individual components is mediated by an alteration to the membrane properties and disruption of the membrane-associated functions at concentrations ranging from 0.06 to 0.6% (*v*/*v*). Its toxicity has been proven against *C. albicans*, *C. glabrata*, and *S. cerevisiae* [84]. Furthermore, tea tree oil has also been recognized as a functional additive to traditional antifungal drugs looking for synergism. A recent proposal recommends hydrogel formulations containing a combination of ketoconazole plus tea tree oil [85].

3.15. Turmeric: (numerous studies have shown that extracts of this yellowish-orange powder obtained from highly branched rhizomes of *Curcuma longa*) display a broad spectrum of pharmacological properties with large medicinal potential [9,86]. Curcumin and other curcuminoids are the specific bioactive components. These compounds are polyphenolic compounds with well-known strong antioxidant and anti-inflammatory properties [87], but also with antibacterial, antiviral, and antifungal activity [88] due to the alteration of the intracellular redox balance and the impairment of the mitochondrial electronic chain transport. The fungicidal effect seems to be especially strong against pathogenic *C. albicans* strains [89]. 

It is obvious that there are a very high number of plant natural extracts currently known. Table 1 includes the most used in folk medicine from ancient times, but many others could be also added, as found at the bottom of the Table [46,60,69,70,90,91,92]. Concerning other possible bioactive components with possible antifungal activity not described in previous oils, triterpenes deserve some comment due to the recent interest in this family. Triterpenes are a large group of phytochemicals that comprise around 20,000 molecules. Due to their hydrophobicity and low solubility, triterpenes were considered inactive for a long time, but recent studies support a wide range of pharmacological capacities. Two intensively studied triterpenes are oleanolic and ursolic acids, but other triterpenes are found in propolis and many herbal extracts. Both acids are bioactive molecules with antimicrobial effects [93], and currently, they are included in the list of compounds with potential applications against human pathogens. Oleanolic acid is mainly found in olive oil, as well as other several other plant species. Ursolic acid is widely found in the peels of fruits like apples, as well as in medicinal herbs like rosemary and thyme or flowers like gladiolus. It seems that triterpenes promote apoptosis by the regulation of mitochondrial function through various signal pathways leading to the suppression of the NFκB pathway and the activation of some caspases. In addition to the possible effect on the fungal cell, oleanolic and ursolic acids are inhibitors of the cellular inflammatory process and of phase 2 xenobiotic bioprocessing enzymes [93].


jof-10-00334-t001_Table 1Table 1Most common NPs endowed with antifungal activity in medical use to combat human mycosis (listed in alphabetical order, but the use of propolis (PP) and carnosic acid (CA) from sage studied by our group are listed in the last two rows). See the text for other details.Natural Extract/Oil(*Scientific Name*)Main Bioactive MoleculesFungal Species TestedConcentrationRef.Aloe Vera gel (*Aloe barbadensis miller*)Aloesin, Aloin*C. albicans*, *A. fumigatus*, *A. niger*, *C. glabrata*, *C. tropicalis*312.5–625 μL/mL extract [52,53]Barberry (*Berberis vulgaris*)Berberine, polyphenols*C. albicans*, *C. krusei*, *C. glabrata*, *C. dubliniensis*, 6 Candida clinical isolates10 μg/mL [54,55]Black pepper oil (*Piper nigrum*); Cayenne pepper (*Capsicum frutescens*)Phenols, terpenes, alkaloids, capsaicinoids*A. flavus*, *A. ochraceus*, *F. graminearum*, *P. viridcatum*, *C. neoformans*, *Candida* spp. 4–16 μg/mL[56,57,58]Cinnamon oil (*Cinnamomum verum*)Cinnamaldehyde*A. niger*, *C. albicans*, *C. auris*25 μg/mL[59,60,61]Clove (*Syzygium aromaticum*) oilEugenol*A. flavus*, *A*, *fumigatus*, *A. niger*, *C. auris*, *C. parapsilosis*, *C. krusei.*5 dermatophyte clinical strains: *Microsporum canis*, *M. gypseum*, *Trichophyton rubrum*, *T. mentagrophytes*, *Epidermophyton floccosum*0.02–20 μL/mL[62,63]Cumin (*Cuminum cyminum*) seed oilCumin aldehyde, cumin quinones*C. albicans*,*Candida* clinical isolates0.02–1.25 μL/mL [64,65]Garlic (*Allium sativum*) extractAllicin, NpRS peptide*C. albicans*,*Candida* clinical isolates12–23 mg/mL (extract)32–128 μg/mL(allicin)[49,66,67,68,69,70]Manuka (*Leptospermum scoparium*) oil and honeySesquiterpenes, β-triketone, acetanisole*Malassezia furfur*, *Trichosporon mucoides*, *C. albicans*, *C. tropicalis*, *C. glabrata*0.01–3.13% (*v*/*v*)[71,72]Maqui Berry (*Aristotelia chilensis*) fruit extractAnthocyanidin (delphinidin) 
*C. albicans*
>10 mg/mL[73,74]Neem (*Azadirachta indica*) extractAzadirachtin*Rhizopus*, *A. flavus*, *A. fumigatus*, *A. niger*, *A. terreus*, *C. albicans*, *Microsporum gypseum*5–20% [75,76]Olive (*Olea europaea*) oilOleuropein, tyrosol, triterpenes (i.e., oleanolic acid) *Candida* spp.,*Saprochaete* spp.,*Candida* clinical isolates,*Saprochaete capitata*50 μL/well(around 10%)[77,78]Oregano (*Origanum vulgare*) oilCarvacrol and thymol*C. neoformans*,*C. laurentii*16–128 μg/mL[46,79,80]Rosemary (*Rosmarinus officinalis*) oil/extractRosmarinic and carnosic acids, terpenes, triterpenes (i.e., ursolic acid)*Alternaria alternata*, *Botrytis cinerea*, *Fusarium oxysporum*, *Aspegillus flavus*50–200 mg/mL,0.5 mg/mL[81,82]Tea Tree (*Melaleuca alternifolia*) oilTerpinenes, acetanisole*C. albicans*, *C. glabrata*, *S. cerevisiae*0.25–1% (*v*/*v*)[83,84,85]Turmeric (*Curcuma longa*) oil/extract
*Curcumin*
*C. albicans*, *C. neoformans*128–256 μg/mL[9,86,87,88]Essential oils/extracts of other sources (thyme, peppermint, anise, camphor, lemon grass, marjoram, caraway, parsley, celery, spinach, onion, coconut, grapefruit, purple coneflower, etc.)Monoterpenes (thymol, carvacrol), sesqui, di and triterpenes, alkaloids, saponins, phenols, flavonoids, organosulfurs, fatty acids, polyphenols, chicoric acid, etc.Mostly *Candida* and *Aspergillus* species. Details at the references indicated (column on the right).Diverse, wide ranges according to the type of oil, the studies, and the fungal strains[46,63,70,90,91,92]Propolis (*Apis mellifera*) resin extractFlavonoids, phenol esters*C. albicans*, *C. dubliniensis*, *C. glabrata*, *C. parapsisolis*, *C. krusei*, *C. tropicalis*, *C. Neoformans*, *S. cerevisiae*4–2000 μg/mL(greatly dependent on geographical area)[41,42,94]Sage (*Salvia officinalis*) leaf extractCarnosic acid, triterpenes*C. albicans*, *C. neoformans*, *C. glabrata*50–500 μg/mL[41,42,95,96]


## 4. Therapies Based on Combinations among Chemotherapy/Natural Products

An interesting therapeutic approach may be the combination of more than one product with antifungal activity, preferentially with different targets, looking for synergistic effects or overcoming possible resistance to one of the mixed components. The worrying increase in acquired resistance shown by common fungal species demands the obtention of new, more potent, safe, and effective antifungal compounds, suitable for the treatment of both superficial and systemic infections. In this way, NPs may not only act in preventive health, but also as true antifungal agents. Several preparations based on mixtures between conventional antifungals and NPs have been tested. They can be classified according to the type of the combined agents.

### 4.1. Combination of Two Antifungal Drugs Used for Chemotherapy

The combination of fluconazole with AmB for preventing the yeast–hyphal transition of *Candida albicans* confirms the potential use of this azole plus polyene mixture against oral candidiasis. In addition to the expected dual respective effects on ergosterol synthesis and binding, an extra specific action of this mixture to account for the synergism observed has been suggested [97], related to the downregulation of the fungal *SAP3* gene. *SAP3* encodes an aspartyl protease involved in the degradation of selected host’s proteins involved in the immune defense, including histatin-5, a peptide from human saliva that possesses fungicidal activity. However, the antifungal effect of AmB is enhanced not only by its combination with fluconazole, since mixtures between AmB and several off-patent drugs have shown additive or synergistic effects against *Cryptococcus neoformans* and *Candida* spp. [98,99].

### 4.2. Combination among Natural Products

Some mixtures of different EOs increase the antifungal activity of each individual oil and reduce the side effects associated with treatments involving fungal chemotherapy. The fungicidal activity of basil, marjoram, clove, cumin, and caraway oils against *C. albicans* and *A. niger* was studied by Hassan et al. [100]. Although clove oil displayed the strongest action, the MICs of the EOs mixtures against both strains were lower than those of any of the individual oils, suggesting a synergistic effect due to the action of several compounds. Moreover, mixtures of purified active components (eugenol, carvone, and cuminaldehyde) displayed higher activity than the combination of crude oils. In summary, the synergistic effect of combinations of EOs or their active components could reinforce the individual antifungal action and decrease side effects. Consequently, they might be used to treat fungal infections caused by the above-mentioned yeast. 

A relevant case of antifungal combinatory treatment based on two natural products widely studied by our research group deals with appropriate mixtures between carnosic acid (CA) and propolis (PP) [41,42]. These NPs are mentioned separately at the bottom rows of Table 1. Briefly, carnosic acid is a natural benzenediol diterpene with antioxidant properties, commercially available due to its wide utilization as a food preservative [96]. This acidic phenolic diterpene is easily extracted in a semipurified form from the leaves of sage (*Salvia officinalis*) and rosemary (*Rosmarinus officinalis*) due to the high content in these sources [95]. In turn, propolis (PP) designs a complex, resinous material produced by honeybees (*Apis mellifera*), whose heterogeneous composition of PP depends on both abiotic and biotic factors and encompasses more than 300 compounds [94]. The preparation of the propolis hydroethanolic extracts, as well as the analytical details of the Chinese propolis used in our studies, is described in [41,42].

Although individually CA and PP show weak antifungal activity [96,101], we found a significant in vitro antifungal activity with specific combinations of both extracts, recording a synergistic action at appropriate CA/PP ratios. Successful results have been obtained against several prevalent fungal pathogens, like *C. albicans* [40], *C. glabrata* [102], *C. neoformans* [41], and *A. fumigatus* [80], and our preliminary evidence from *A. flavus* was also satisfactory. Data from *Candida* spp. point out a strong induction of intracellular oxidative stress harmful to fungal metabolism. However, in the case of *C. neoformans*, PP exerts the antifungal effect by another supplemental mechanism, as it has been reported that PP diminishes the content of chitin and melanin production involved in cell wall integrity, reducing *C. neoformans* virulence [103].

Concerning the use of CA + PP mixtures in the treatment of clinical candidiasis, (i) the mixture has been successfully tested in vitro against an oral clinical isolate of *C. albicans* from a patient at the Hospital “La Fe”, Valencia, Spain; (ii) a preliminary clinical trial carried out with chosen patients affected at different stages of oral candidiasis supports the therapeutic validity of the CA + PP formulation studied. These data are industrial property of the company Vitalgaia, S.A., according to a study approved by the Research Ethics Committee of HM-Hospitals, Spain (Ethic code: 16.06.0961GHM). However, some preliminary data have been published [104]. 

### 4.3. Combination of Antifungal Drugs and Natural Products

It seems reasonable that the most potent approach to combat mycotic infections should be the elaboration of new formulations between established antifungal drugs and NPs. This approach looks for synergy between both agents, presumably directed against different targets to achieve a double disruption of fungal metabolism. Thus, EOs and their combinations with antifungal drugs may provide useful options for skin disinfection and the treatment of *Candida* and other fungal infections. Thus, an in vitro synergistic action between Polish propolis and two azoles (fluconazole and voriconazole) was effective in the eradication of preformed biofilms [105]. The combination of carvacrol with fluconazole or AmB against *C. albicans* has also been used with good additive results [79]. Another study reported the combination of AmB with eugenol or with a set of off-patent drugs, which was effective against several pathogenic yeasts [99,106]. Testing a different type of NP, the flavonoids quercetin and rutin induced antifungal synergy when supplied in combination with AmB against *Candida* spp. and *Cryptococcus neoformans* [107].

On the other hand, some formulations do not produce a synergistic or even additive antifungal effect. For instance, the efficacy of formulations comprised of three EOs (lemongrass, clove, and cinnamon) in combination with fluconazole, AmB, flucytosine, and micafungin were explored [63]. The obtained results were surprisingly diverse. While synergism was never observed with cinnamon oil or any of the antifungal drugs, lemongrass oil displayed synergistic, additive, and indifferent interactions depending on the selected antifungal drug. Furthermore, combinations of clove oil with fluconazole and flucytosine gave rise to synergistic interactions, but surprisingly the interaction with AmB was antagonistic. Remarkably, mixtures of PP plus specific antibacterial or antifungal antibiotics seem to induce cooperative activities [108]. Related to our experience in PP combinations, preliminary experiments with an oral isolate of *C. albicans* revealed that preincubation with CA:PP was able to further increase the lethal activity of AmB [41]. However, additional research is necessary, since the strong toxic effect produced by just AmB might mask the results obtained with the CA:PP mixture plus the polyene. 

## 5. Some Pending Questions Regarding Benefits and Drawbacks of NPs as Alternative in Antifungal Therapy

As stated above, some bioactive plant components applied as beneficial remedies in folk medicine from ancient times have already been incorporated into antimicrobial therapy affording promising expectations. However, in our view, these hopeful lights are still accompanied by considerable shadows. Some pending points must be properly addressed before this strategy can be definitively introduced with efficiency and reliability. Likewise, although notable antifungal actions have been attributed to many plant extracts and EOs, the efficacy of those actions depends on several unpredictable factors. We will summarize the main points that should be addressed.

Many NPs are complex mixtures of numerous components, so it is not feasible to determine which are the true bioactive molecules responsible for the recorded antimicrobial effects. It seems evident that the actual activities shown by many tested NPs are not clearly determined. For some compounds, the scientific evidence is weak, and data regarding their safety and efficacy are controversial. Therefore, a complete analysis of noteworthy natural extracts and identification of those essential components must be a priority task. This analysis would allow for improving future natural formulations. In the particular case of using carnosic acid (CA) and propolis (PP), a successful combination has been deeply studied. CA is a well-defined component easily extracted from sage or rosemary leaves by reproducible methods, but raw PP is a material with a highly heterogeneous composition. As previously reported, we have found that batches of PP from different countries display very different antifungal capacities.

It is usually assumed that NPs do not produce damage in hosts, but this possibility cannot be ruled out. NPs are not always a synonym for better and healthy action. Likewise, inert, or even toxic agents might also be present, i.e., camphor and several alkaloids. Some NPs contain specific components with proven antifungal activity, but consistent data support that NPs are essentially beneficial as antioxidants and immunomodulators. Indeed, distinct reports point out that their immune-boosting properties can be responsible for the health effects as adjuvants rather than through their direct fungal toxicity. However, they can also trigger unforeseen harmful effects or interactions with incompatible medications. 

Except in certain cases, due to their normally weak activity, they are unable to ensure a complete cure of systemic mycotic infection, mainly affording supportive or complementary help. It should be considered that the conditions established at the laboratory to test their antifungal activity could differ from those occurring in the case of an infection. The presumed recorded in vitro effects may be absent or just different in infected patients.

Assays involving clinical trials performed with isolates of pathogenic fungi responsible for outbreaks in hospitals and the community are scarce. The possible beneficial effects of NPs have been tested in a considerable number of clinical trials, but most of them focused on cancer treatments to improve the side effects of aggressive anticancer therapies. Green tea, grape seed, pomegranate, ginger, ginseng root, or broccoli have been the most used extracts, and curcumin, resveratrol, gallates, and epigallocatechin are the most tested NPs [109]. Concerning clinical trials looking for antifungal action, the database Clinicaltrials.gov contains only four registered studies regarding NPs [110,111,112,113] versus about one hundred related to the use of classical clinical antifungals, polyenes, azoles, and echinocandins. Unfortunately, the potential usefulness found in those four trials is generally small and still remains elusive. Likely, the reason for those scarce results comes from the fact that they have been tested individually instead of as adjuvants of other medications. 

## 6. Concluding Remarks and Perspectives

The evidence collated in this review allows for the conclusion that clinical therapy against the most common pathogenic fungal outbreaks displays serious challenges that demand urgent and more efficient responses. The main limitations come from the narrow arsenal currently available for chemotherapy. In addition, the used drugs are frequently harmful to patients, causing important unwanted effects. Furthermore, a significant number of innocuous fungi have recently turned out to be virulent, a fact joined by the worrying increase in acquired resistance against conventional drugs.

In the context of searching for new, safer, and more potent antifungal treatments, NPs are now being proposed as an alternative or complementary approach to current chemotherapy. A great variety of NPs have historically been part of preventive health practices. Many of them exert a moderate fungistatic or fungicidal action through mechanisms still partially unknown. A detailed analysis of the accessible data regarding MIC, toxicity, resistance, reliability, and related parameters indicates that NPs endowed with higher antifungal activity are clove and cinnamon oils, fresh garlic extract, and CA/PP mixtures. However, for the majority of NPs, the precise identification and characterization of the true bioactive principles responsible for the antifungal properties contained in each specific compound remains a research priority. Another crucial factor for the validation of NPs as useful antimycotics must be a substantial increase in clinical trials, expanding both the number of compounds tested and the spectrum of potentially pathogenic fungal strains, and establishing actual scenarios of infections rather than laboratory conditions. Nevertheless, there is ample consensus on the weaker fungicidal activity of NPs in comparison to the compounds used in clinical therapy.

However, a comparison of NP doses required for antifungal activity in several cases showed a range like that established for chemotherapeutical drugs. Thus, NPs could become a convenient substitute treatment, minimizing undesirable effects. According to the MIC values, the order of most active NPs would be as follows: barberry, whose MIC range is as low as 10 μg/mL [54]; CAY-1, the bioactive saponin found in cayenne pepper, shows a MIC range from 4 to 16 μg/mL [58]; and cinnamon oil also has a relatively low MIC value, 25 μg/mL [59]. Finally, data from cumin oil present a similar low but wide range of MIC, from 0.02 to 1.25 μL/mL, depending on the *Candida* strain tested [64] and the eugenol content of the batch. Similar variability can be observed in garlic oil as a function of their allicin content. Knowing the density of these oils would be necessary for comparison with other data, due to the different units reported in the referred cumin study. On the other hand, maqui berry, rosemary, and aloe vera extracts show low antifungal activity, since the concentrations required for obtaining a noticeable effect are more than 1000 times higher, around >10 mg/mL, 50–200 mg/mL, and around 625 μL/mL, respectively [53,74,81,82]. These NPs should be considered as adjuvants for accompanying other antifungal agents rather than as true alternative options for exclusive treatment. This idea agrees with our findings concerning carnosic acid from rosemary or salvia extracts as a complement to propolis [41,42].

In this context, we envisage the main application of NPs in combinatory approaches looking for synergy, because the feasible interactions between two fungistatic or fungicidal agents might enhance the mere sum of the respective single activities, although neutralization or at least non-additive reactions cannot be excluded. In this way, some CA + PP formulations display a remarkable synergism becoming a successful therapeutical tool comparable to conventional antifungals. Interestingly, the activator effect could even be higher when that mixture is further combined with AmB, which induces a significant reduction in the number of polyene doses, therefore ameliorating its adverse effects. 

In turn, in possible assays with EOs containing eucalyptol and flavonoids, the reported double opposite effect of the monoterpene and the flavonoid on the expression of *ERG11* and *CDR* fungal genes can give place to neutralization rather than an additive effect. Therefore, an important conclusion points to whatever proposed combination should be analyzed with caution before use, in order to avoid noxious consequences.

We are certainly convinced that rigorous NP studies afford promising perspectives and may become a valuable and wealthy alternative for antifungal chemotherapy in the near future. It is likely that artificial intelligence and mathematical modeling will include algorithms that would be beneficial for designing combination treatment protocols, choosing the nature of the chemotherapy agent according to possible resistance in the fungal strain responsible for the infection, and the most appropriate NP accompanying or even replacing conventional chemotherapy in specific infections. The reduction in doses or shortening of the period of treatment with the chemotherapy agent because of the addition of innocuous NPs would also be parameters that might be optimized by artificial intelligence and mathematical modeling in the near future. However, more experimental data, in vitro or obtained in clinical trials, should be collected before applying those powerful tools in common medical practice. Currently, we think that, except for some confirmed exceptions, the principal utility of NPs so far must be directed to reinforce and improve chemotherapy antifungal drugs rather than replace them. 

## Figures and Tables

**Figure 1 jof-10-00334-f001:**
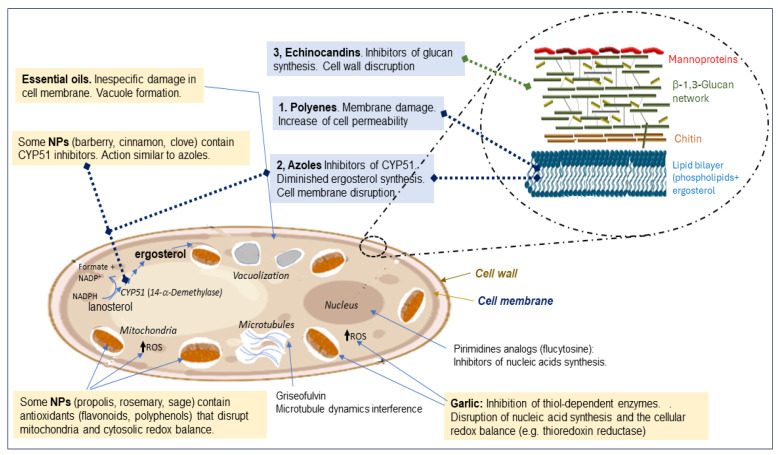
Scheme of a fungal cell showing the cellular targets for the main three families utilized in antifungal chemotherapy (blue background) and for best studied NPs (light yellow background). The corresponding mechanisms of action are outlined. Dashed lines indicate inhibition.

## Data Availability

Not applicable.

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
