# Peer review of "Natural Substances as Valuable Alternative for Improving Conventional Antifungal Chemotherapy: Lights and Shadows"

_jof, 2024, doi:10.3390/jof10050334_

Round 1

Reviewer 1 Report

The manuscript ''Natural substances as valuable alternative for improving conventional antifungal chemotherapy: Lights and Shadows'', by Argüelles JC et al. approaches an important subject in these days - the antifungal resistance, especially after the fungi were introduced in the pathogens list of WHO.

The manuscript has a logical flow. However I suggest the following changes.

- line 10 - what do you mean by ``A single Fungi are..`` ?

- line 50 - please check number if deaths

- line 56 - correct the percent of people dying of candidiasis

- figure 1 - I would suggest moving it below, after chapter 3, after you have also discussed the natural products

- line 292 - ''This is a t Most used'' - please correct

- table 1 - move it either before or after enumerating all the subsection with different compounds.

- line 640 - please rephrase ''After this revision''

In addition please emphasis the mechanisms of antifungals and of e few of the natural compounds.

Please discuss the dose required for some of the general compounds.

Also approach and cite some papers with in vivo experiments on the effect of natural compounds of antifungals.

Add a section on how artificial intelligence and mathematical modelling may help in finding new molecules.

Reviewer 2 Report

Improve the first part of the introduction. In the attached file I mention each of the details.

They wanted to address many natural compounds, but I think they could choose the most important ones and describe them better.

I consider that a review  is to address a topic of interest in depth. I think that here all the possible natural treatments are being approached in a very superficial way.

Modify the order so that the text is more understandable.

Organize the format of references from 1 to 9.

It is a topic of interest, however, I consider that the way in which it was addressed is not correct. The mechanisms of action of these natural products against the different species of fungi are not described in detail. Furthermore, the information they give about each one is very simple. That's why I think it should be restructured, choose some of the most relevant natural products and focus on those. They talk about fungal species of clinical importance but only focus their work on Candida.

Each of the errors found are underlined in the attached file.

Reviewer 3 Report

In this manuscript, the authors review the current state of mycoses therapy and the effect of using natural substances as adjuvant or replacement therapy. Natural substances mean herbal remedies of uncertain composition, historically used in folk medicine, as well as propolis and honey.

The topic of the review is certainly very important. At the present stage, there is an increase in mycoses, the spread of resistant forms of pathogens, as well as an expansion of the range of fungal pathogens. As a result, there is an increase in deaths, which is also facilitated by a general decline in the population’s immunity. Against this background, the use of natural substances is certainly useful.

The authors provide general data on such therapy, and also specifically consider the properties of individual natural substances. They emphasize the importance of analyzing the entire composition of plant concentrates and identifying active components, the problem of compatibility of biologically active substances, and noted the different quality of raw materials obtained from different territorial sources. This direction of research contributes to the standardization of herbal medicines, which, in the authors’ terminology, certainly represents “light.”

The advantage of the manuscript is also a complete assessment of information on individual natural compounds. For specialists immersed in the problem of mycoses, information on “shadows” is no less important, which is not hushed up by the authors.

I think the review is very useful and timely.

From the text versions sent to me, it is clear that work has been done to improve the manuscript. I believe that it is advisable to publish it without further changes (review-v2).

This is a difficult topic to cover, but you did a good job.

Author Response

See attached reply.

Best wishes, 

Reviewer 4 Report

Dear authors,

I have read the prepared manuscript and the first thing I have to point put is that it is too general. Similar reviews already exists and the novelty of presenting the results in the available literature is missing. Also, figure is too complex and on the other hand the table provide minimum information.

Missing reference for the statement in lines 34-35

Missing reference for the statement in lines 47-48

Figure 1. There is too much text. Try to make it simpler, but to keep presented information

Table 1. This table is too simple. It does not give enough information. Please, incorporate columns related to the microorganisms (or just fungi) sensitive to the NPs, concentration of NPs and if applicable mechanism of action

Author Response

See attached document with the point-to-point replies.

Best wishes

Reviewer 5 Report

The manuscript provides an overview of known natural compounds to be used in combination with conventional therapy in local and systemic fungal infections. The authors covered several aspects of natural products active on fungi: their chemical class, the mechanism of action, undesirable effects and toxicity. The opportunity of combining antifungals from natural source with synthetic compounds to achieve a resolution in the case of severe infections was also highlighted. The authors also evidenced the positive and critical aspects in associations with synthetic drugs.

The manuscript is clear and well organised. The novelty and originality of this manuscript is good .The data reported are well supported by the references cited.

However, it needs some improvements before its acceptance.

Figure and Tables are well presented .

The References  are adequate for the characteristics of the manuscript

The manuscript can be accepted after minor revisions.

                        Some specific comments are listed below:

Page 4 line 158 : add a reference

Page 4 line 181-182… Those synthetic compounds…: the sentence refers to the compounds just mentioned, but griseofulvin is a natural compound. Perhaps it refers to synthetic derivatives? Rephrase the sentence (see https://doi.org/10.1021/acs.jafc.2c09037)

Page 6 line 243: add the reference

Page 6 line 249 : the reference 45 is not appropriate, please check.

Page 6 line 252: the reference 48 is not appropriate, please check.

Page 6 line 258: the reference 50 is not appropriate, please check

Page 6 line 291: add the reference.

Page 7 line 298: add the reference

Page 9 line 433: add the reference

Author Response

See attached document with the point-to-point replies

Round 2

Reviewer 1 Report

The authors have answered all my recommendations.

 I would like to thank the authors for their detailed replies and for their kindness in answering.

Author Response

Thank you!

Reviewer 2 Report

check the spelling of the scientific names of the species.

The comments mentioned in the first review were addressed.

Author Response

Thank you!
